chemical engineering/environmental engineering/analytical chemistry

heavy metal, statistical classification, light scattering, polystyrene, support vector machines

**Author for correspondence:**
Jeong-Yeol Yoon
e-mail: jyyoon@email.arizona.edu

†These authors contributed equally to this study.

This article has been edited by the Royal Society of Chemistry, including the commissioning, peer review process and editorial aspects up to the point of acceptance.

# Mie scattering and microparticle-based characterization of heavy metal ions and classification by statistical inference methods

Katherine E. Klug[1,†], Christian M. Jennings[2,†], Nicholas Lytal[3], Lingling An[1,3,4] and Jeong-Yeol Yoon[1,2]

[1]Department of Biosystems Engineering, [2]Department of Biomedical Engineering, [3]Statistics Graduate Interdisciplinary Program, and [4]Department of Biostatistics and Epidemiology, The University of Arizona, Tucson, AZ 85721, USA

NL, 0000-0003-2134-2715; LA, 0000-0001-8273-0776; J-YY, 0000-0002-9720-6472

A straightforward method for classifying heavy metal ions in water is proposed using statistical classification and clustering techniques from non-specific microparticle scattering data. A set of carboxylated polystyrene microparticles of sizes 0.91, 0.75 and 0.40 µm was mixed with the solutions of nine heavy metal ions and two control cations, and scattering measurements were collected at two angles optimized for scattering from non-aggregated and aggregated particles. Classification of these observations was conducted and compared among several machine learning techniques, including linear discriminant analysis, support vector machine analysis, K-means clustering and K-medians clustering. This study found the highest classification accuracy using the linear discriminant and support vector machine analysis, each reporting high classification rates for heavy metal ions with respect to the model. This may be attributed to moderate correlation between detection angle and particle size. These classification models provide reasonable discrimination between most ion species, with the highest distinction seen for Pb(II), Cd(II), Ni(II) and Co(II), followed by Fe(II) and Fe(III), potentially due to its known sorption with carboxyl groups. The support vector machine analysis was also applied to three different mixture solutions representing leaching from pipes and mine tailings, and showed good correlation with single-species data, specifically

with Pb(II) and Ni(II). With more expansive training data and further processing, this method shows promise for low-cost and portable heavy metal identification and sensing.

# 1. Introduction

Heavy metals in contaminated water and soils are an important focal point in environmental regulation and monitoring due to their human and environmental health risks [1,2]. Conventional methods for monitoring heavy metals use atomic absorption/emission spectroscopy (AAS/AES) or ion-selective electrodes (ISE). AAS/AES techniques broadly identify heavy metal ions based on their spectral fingerprints, whereas ISE techniques identify single heavy metal ions based on their specific activity with size-specific molecular cavities in an electrode material [3–5]. However, these both face challenges in portable or on-site monitoring applications. AAS/AES analytical techniques require that samples be transported back to a central laboratory facility for testing, as these instruments are large and require significant infrastructure, data processing and technical expertise for use. ISE analytical techniques are generally portable; however, they face different limitations attributed to electrode fouling/interference and loss of sensitivity during use [6]. Additionally, the single-species specificity of ISE devices prevents their use in identifying and quantifying diverse heavy metal species in a single assay. In the light of these challenges, there is interest in developing portable sensing techniques that can be used to classify a wide range of heavy metal species by spectral or electrochemical fingerprint. Towards these goals and other removal processes, researchers have explored the use of functional groups involved in biosorptive metal uptake, including carboxyl groups, as specific chelation agents [7–10].

In this work, we propose a novel technique for classifying heavy metals using Mie scattering measurements associated with each ion's characteristic activity with different sizes of carboxylated polystyrene microparticles (PS-COOH). While superior specificity could be achieved through using functional groups specific to each heavy metal ion species, covalent conjugation of such functional groups to polystyrene particles is not always easy and may deprive the particles of their stability. Carboxylated particles, on the other hand, are readily available commercially with relatively low cost, and very stable for a long period. Based on our own observations, particles would not precipitate out from suspension for more than 1 year. The main objective of this work is an exploration of how well statistical inference methods could classify the heavy metal ion species in the water samples using non-specific assays, e.g. with carboxyl groups.

Light scattering from these particles falls in the Mie scattering regime, as the wavelength of incident light (blue; 460 nm) is comparable to the size of particles (0.4, 0.75 and 0.91 μm). Ion species of differing atomic mass and oxidation states are expected to interact characteristically with these carboxyl groups on differently sized microparticles. Furthermore, the extent of inter-particle aggregation is expected to vary by the heavy metal ion species and their concentrations. Such particle aggregation significantly alters the extent of Mie scattering, when measured at an optimized detection angle, as Mie scattering changes significantly over the particle size and scattering angle (figure 1). We hypothesized that heavy metal ions may be characteristically classified through pattern recognition and statistical inference techniques applied to large multivariate scattering datasets collected across different particle sizes and detection angles. Our initial findings demonstrate promising single-species identification and quantification by multivariate pattern recognition analysis. When fully developed for mixed species samples, which requires vast amounts of data covering all possible concentration combinations of binary and tertiary mixtures, this method may provide the broad multi-species detection advantages of AAS/AES techniques in a portable format for versatile and on-site heavy metals detection.

# 2. Material and methods

## 2.1. Reagent preparation

Separate heavy metal stock solutions were prepared in deionized water at concentrations between 1000 and 300 ppm (mg l$^{-1}$). Lead nitrate [Pb(II)], zinc sulfate heptahydrate [Zn(II)], copper sulfate pentahydrate [Cu(II)], chromium trioxide [Cr(VI)], potassium chloride (K$^+$) and magnesium chloride hexahydrate (Mg$^{2+}$) were acquired from Sigma-Aldrich (St. Louis, MO, USA). A cadmium reference standard [Cd(II); 1000 mg l$^{-1}$ ± 1%, 2% HNO$_3$], nickel chloride hexahydrate [Ni(II)], cobalt chloride

**Figure 1.** (a) Mie scattering intensity is collected at a specific detection angle through inserting a detection optical fibre into one of the holes in the 3D printed device. The angle showing the maximum Mie scatter intensity shifts to a different angle upon particle aggregation, which can be used for identifying heavy metal ions. (b) Carboxylated polystyrene microparticles characteristically interact with heavy metal ions ($Me^{2+}$). This leads to particle aggregation, e.g. effective particle size changes from singlets (green) to aggregates (red), seen here with 0.91 μm particles in the presence of 1 and 10 ppb Cr(VI). These shifts can be quantified through angle-specific scattering measurements.

hexahydrate [Co(II)] and iron chloride hexahydrate [Fe(III)] were acquired from Thermo Fisher Scientific (Waltham, MA, USA). Iron chloride tetrahydrate [Fe(II)] was acquired from Honeywell (Morris Plains, NJ, USA). From these stock solutions, assay solutions of 2000–2 ppb (ng l$^{-1}$) were prepared through serial dilution in deionized water along with a separate blank sample (0 ppb). Surfactant free, spherical carboxylated polystyrene microspheres (PS-COOH) of diameters $D = 0.91$, 0.75 and 0.40 μm (10% w/v; surface density of carboxyl group = 9.17 Å$^2$/COOH) were purchased from Magsphere, Inc. (Pasadena, CA, USA). These stocks were diluted to 0.03% w/v samples for all assays before being mixed with equal volumes of heavy metal ion solutions.

## 2.2. Mie scattering simulations

Mie scattering simulations were conducted using MiePlot v4.6 (Philip Laven, www.philiplaven.com/mieplot.htm) to determine appropriate forward scatter detection angles for each particle size, considering singlets (sphere diameter = $D$) and aggregates (sphere diameter ≈ $2D$) [11,12]. These angles were determined to be 90°/50° for 0.40 μm singlets/aggregates, 50°/20° for 0.75 μm singlets/aggregates and 50°/30° for 0.91 μm singlets/aggregates (figure 2). The simulation assumed the refractive index of sample medium (water) = 1.33, the refractive index of polystyrene particle = 1.5725, a 460 nm incident light and a concentration of $2 \times 10^{-4}$–$10^{-5}$ particles/μm$^3$ (corresponding to 0.015% w/v, one half of 0.03% w/v). Detection angle axes were established such that 90° corresponded to scattering perpendicular to the incident light source, 0° corresponded to the direction of forward transmitted light and 180° back to the incident light source.

## 2.3. Single-species Mie scattering assays

Based on scattering simulations and empirical confirmation, the two angles offering the highest scattering peak distinct from the transmitted light for particle singlets and aggregates were used to fix the reflection probes during scattering detection. Assay samples containing single heavy metal species or single reference cation species (K$^+$ or Mg$^{2+}$; chosen from groups 1 and 2 from the periodic table considering their atomic mass and environmental availability) were prepared separately with each PS-COOH sample ($D = 0.91$, 0.75 and 0.40 μm) in a 50 : 50 ratio. This resulted in assay samples containing 0.015% w/v PS-COOH and mixed separately with 1000, 100, 10, 1 and 0 ppb concentrations of each heavy metal species. Aliquots of 1.5 ml were placed in cleaned cylindrical glass cuvettes and inserted into a 3D printed cuvette holder designed in SolidWorks (Dassault Systèmes,

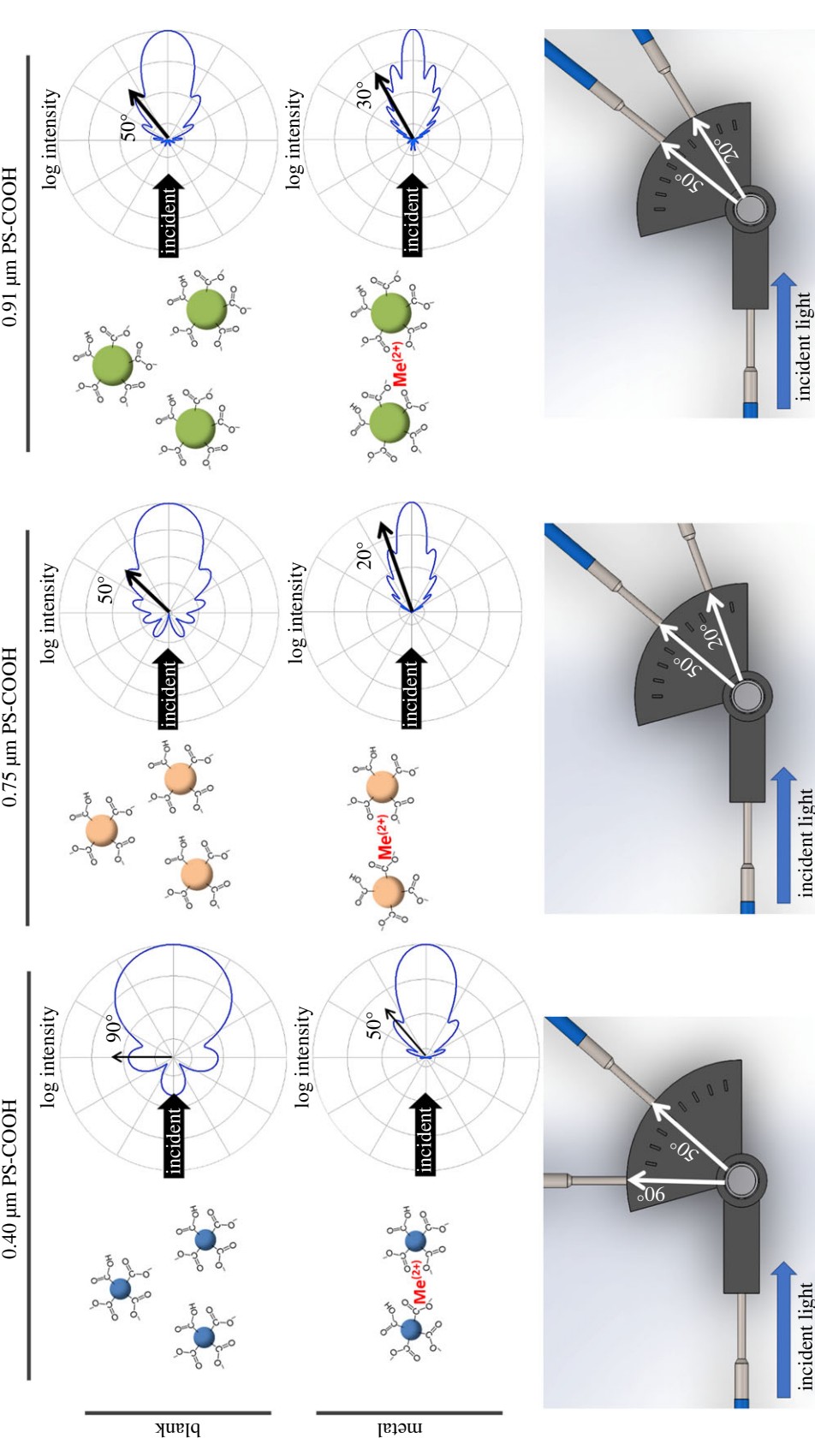

**Figure 2.** Mie scattering simulations and determination of the optimal scattering detection angles for 0.40 μm, 0.75 μm and 0.91 μm partide singlets and aggregates. Scattering measurements were collected using a spectrometer at each of the above angles for heavy metals mixed with the defined particle samples.

Vèlizy-Villacoublay, France) and printed using a Makerbot Z18 3D printer (Makerbot, New York, NY, USA). An incident light of 460 nm was supplied and reflection probes were fixed at the two forward scatter angles determined by the above simulations. Scattering intensities were measured using a USB4000 miniature spectrometer running OceanView version 1.4.1 software (Ocean Optics, Inc, Dunedin, FL, USA). Relative scattering intensity was calculated as $I/I_0$ at each detection angle for each particle and metal species/concentration combination. I is the peak scattering intensity from the given 0.015% PS-COOH particles and metal solutions (1000, 100, 10 and 1 ppb), and $I_0$ is the same measurement from the particles and blank sample (0 ppb). Assays were conducted in triplicates, each time using different samples.

## 2.4. Mixed species Mie scattering assays

Three environmentally relevant mixed target solutions were prepared. The first two simulated pipe leaching into drinking water, specifically Pb(II) and Cu(II) [13,14]. The concentration combinations specifically followed the U.S. Environmental Protection Agency (EPA) mandated action levels (15 ppb Pb(II) + 1300 ppb Cu(II); referred to as 'Pipe 1') and the criteria for reduced tap monitoring (5 ppb Pb(II) + 650 ppb Cu(II); referred to as 'Pipe 2'), as laid out in the Lead and Copper Rule [15]. The third mixture was used to simulate subsurface water contamination in mine tailings (190 ppb Cd(II) + 110 ppb Ni(II); referred to as 'Mine'), as inferred from investigations by Schaider *et al.* [16] and Adamu *et al.* [17]. Each mixture was prepared at twice the desired concentration, and samples were mixed with each PS-COOH sample in a 50 : 50 ratio. As with the single-species assays, relative scattering was calculated from the peak scatter intensity relative to a blank sample at each angle for each particle size.

## 2.5. Multivariate statistical analyses for classification

Using the above scattering data under each particle size and associated detection angle condition, we developed explicit classification functions for each heavy metal and reference cation species across all concentrations through linear discriminant analysis (LDA) based on Mahalanobis distance criteria. These groupings were compared with those provided by K-medians and K-means cluster analyses following Euclidean distance criteria ($k = 11$ clusters). The resulting models' performances were assessed by leave-one-out cross-validation. Following classification of the heavy metal and cation species, we developed separate classification models to predict concentration values. Independence of observations was assumed based on sampling structure. Assumptions of multivariate normality were evaluated graphically through quantile–quantile plots, while assumptions of multicollinearity and homoscedasticity were evaluated through pairwise correlation analysis ($|r| < 0.8$) and graphical analyses.

We also used support vector machines (SVM), a supervised learning method, when categorizing each heavy metal species. For SVM, we divided the data into a training set containing two randomly selected replicates from each particle size and concentration, and a test set consisting of the remaining replicate for each particle size and concentration. The training data, complete with labels for each category, were used to create a classification model based on other parameters within the data. In the case of our data, the categories are the heavy metals, and the parameters on which our model is based are the particle refraction indices for the scattering angles. For test data subjected to SVM, we assessed the correct classification rate of predicted group to actual group using the resulting Cohen's kappa coefficient [18]. Due to the stochastic nature of SVM, we executed the procedure several times to assess the range of Cohen's kappa statistic across all trials, taking 30 random samples for single-species classification and 120 random samples for single-species concentration classification. We also analyzed the effectiveness of SVM models with *t*-distributed stochastic neighbour embedding (t-SNE) plots, a nonlinear dimension reduction method able to visualize/detect patterns that principal component analysis (PCA) may miss [19].

# 3. Results and discussion

## 3.1. Mie scattering assays of single-species samples

Mie scattering assays were first conducted with single species of heavy metals mixed with the three sizes of PS-COOH particles, with measurements at two angles for each particle size – the first specific to

particle singlets (i.e. non-aggregated) and the second specific to particle aggregates. Mie scattering describes the static light scattering regime for spherical particles in which particle diameter is approximately equal to wavelength (between approx. $\lambda/10$ and $10\lambda$, or $50$ nm and $100$ µm for visible wavelengths). Over this range, scattering intensity oscillates as a function of scattering angle, as shown in figure 2. Such scattering patterns are varied by the particle size, refractive index and light wavelength. Given that the refractive index and light wavelength are fixed in these experiments, the particle size remains the major contributor among other parameters [20]. As this relationship changes with increasing particle size, changes in scattering intensity detected from a fixed angle can be used as a proxy measurement for quantifying the extent of particle aggregation.

Based on these initial analyses, the assays using $0.40$ µm particles showed a general decrease in scattering with increasing heavy metals concentrations, at both $90°$ and $50°$ detection angles (figure 3, top). This trend was most strongly evident for Pb(II), Zn(II), Ni(II), Fe(II) and Cr(VI), while Cd(II), Co(II) and the control cations ($K^+$ and $Mg^{2+}$) showed the least significant change. The assays using $0.75$ µm and $0.91$ µm particles showed a similar trend, although the results are different from each other for certain ion species, e.g. Zn(II), Ni(II), Fe(II), Fe(III) and control cations (figure 3). We speculated that heavy metal ions would interact differently with carboxylated polystyrene particles than with alkaline earth metal ions ($Me^{2+}$; $Mg^{2+}$ was evaluated in this work) due to differences in electronegativity and atomic size. Stronger electronegativity and bigger atomic size would lead to larger aggregations of carboxylated particles. While no singular distinguishing characteristic was immediately apparent from the raw scattering measurements, multivariate differences did emerge. These were primarily classified through LDA as well as SVM, described in the following section.

## 3.2. Single-species classification by LDA and SVM

Discriminant analysis is a generative modelling technique in which observations are classified according to an algorithm developed around pattern features extracted from multivariate data [21]. In contrast with PCA, which extracts features for describing data by reducing the observational variance across all data dimensions without formal description of class structure, linear discriminant analysis (LDA) with Bayes Discriminant Rule explicitly models inter- and intra-class variance through Bayesian classifiers following normality assumptions within each class [22,23]. When LDA modelling is used with a diverse, multivariate training data set, the result is a set of classification functions that may be applied for grouping new observations. For small samples of high dimensionality (e.g. this work), LDA can be more suitable for developing classification and predictive models than regression techniques for differentiating future observations [24,25].

Based on a full LDA model developed using all scattering data available at all angles for all particle sizes and concentrations, we found reasonable tentative classification capability ($\geq80\%$ positive classification during leave-one-out cross-validation) for Pb(II), Cd(II), Ni(II), Co(II), Fe(III) and $K^+$ (figure 4 and table 1). This was supported by the distinction between clusters for observations plotted on the first two discriminant functions. From the variable loading plot, relationships are apparent between some ion species and scattering data, for example, between Ni(II) ions and scattering from $0.91$ µm particles at $50°$. The underlying mechanisms of these relationships are undefined as of yet, but they may be related to other findings of specific Ni(II) uptake in carboxyl groups [26].

Additionally, SVM is a discriminative modelling technique where observations are classified according to their location relative to a decision boundary. In contrast with LDA, which is particularly useful for modelling linear relationships, SVM can be used to classify nonlinear data [27]. Specifically, SVM manages this by mapping training data into higher dimension kernel space that can then be separated by a linear hyper-plane [28]. Unclassified observations can then be applied to the trained SVM model and will be classified as one of the classes that the SVM model was trained on. In the case of multiple classes, as with this data, SVM trains multiple binary classifiers and uses a voting scheme to determine the appropriate class.

An SVM model trained on two-thirds of all scattering data available at all angles for all particle sizes and concentrations was tested using the remaining one-third of the dataset (figure 5). We found the SVM classification model had a Cohen's kappa coefficient of $78.67\%$ over 30 random samples with control cations included and $89.26\%$ over 30 random samples with control cations excluded in model. The SVM model was effective (greater than or equal to $80\%$ positive classification) at classifying Pb(II), Cd(II), Ni(II), Co(II), Fe(II) and Fe(III). Over $90\%$ positive classifications were achieved for Pb(II), Cd(II), Ni(II) and Co(II). These positive classification values include all concentration data, including 1 ppb.

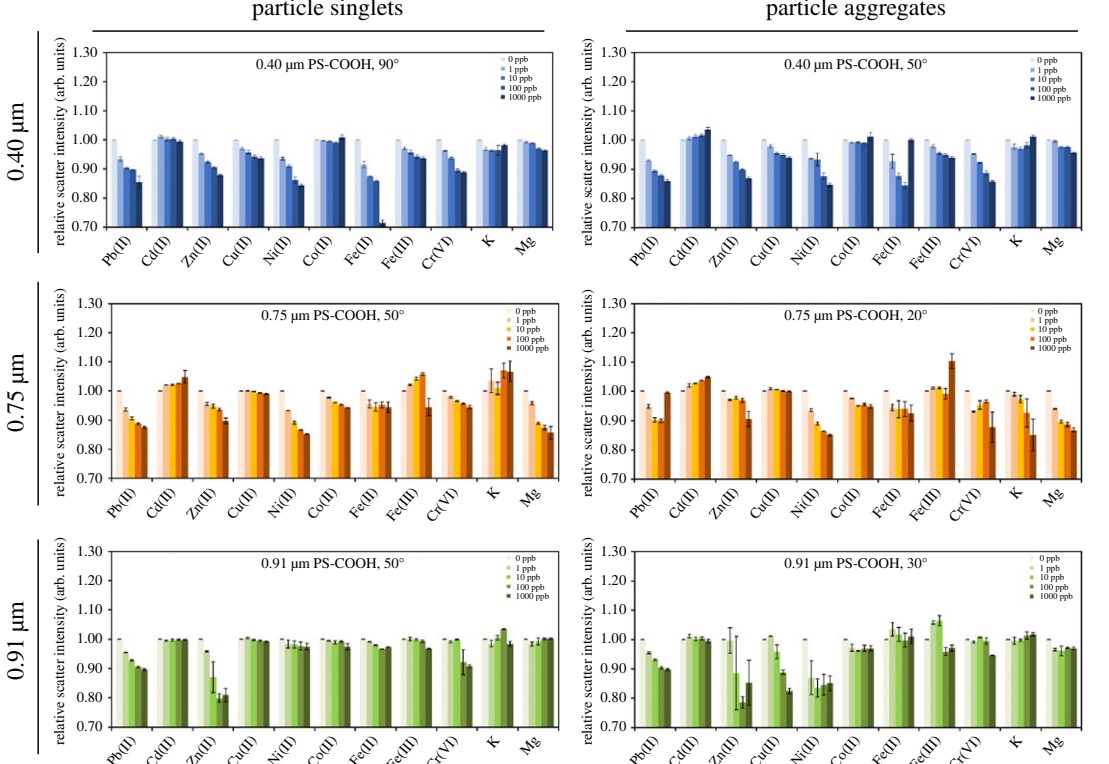

**Figure 3.** Mie scattering data for heavy metal ion and control cation solutions at the defined angles and in mixtures with 0.40 μm, 0.75 μm and 0.91 μm PS-COOH particles. Ions are sorted in a descending order of atomic mass. Averages from three different experiments. Error bars represent standard error.

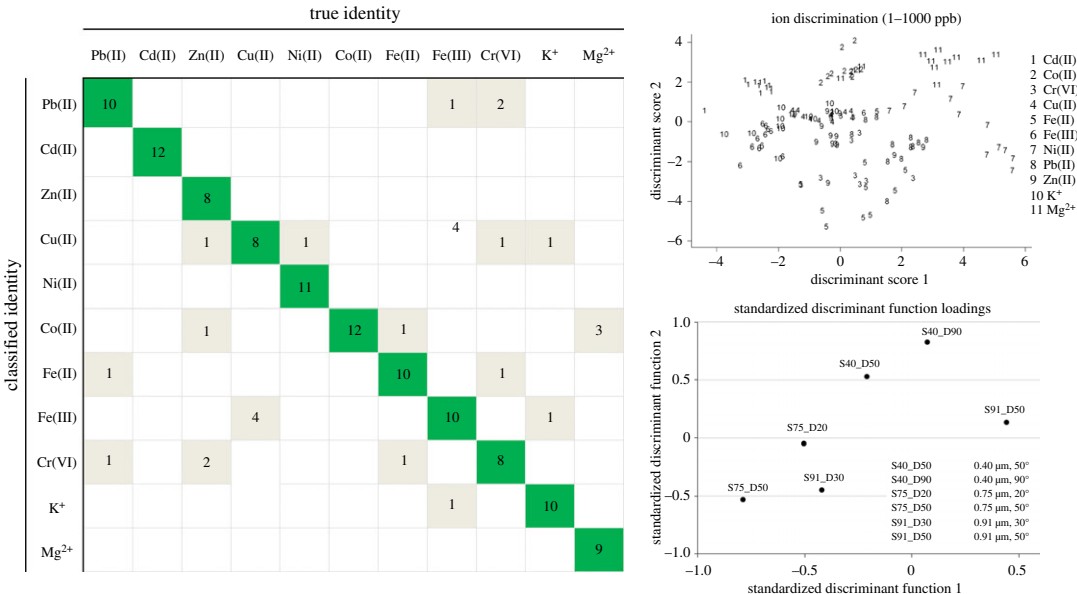

**Figure 4.** Heavy metal ion and control cation classification by linear discriminant analysis (LDA) of all scattering observations at all scattering angles and particles sizes for all ion concentrations. Leave-one-out K-fold cross-validation results and observation clusters plotted on the first two discriminant score functions show reasonable classification capabilities for a small dataset. Standardized discriminant function loadings show the contributions of each variable in resolving these clusters. Ions are organized from highest to lowest atomic mass.

The LDA and SVM models were also compared with K-means and K-medians cluster analyses. K-means and K-medians cluster analysis routines are similar to LDA in that they group observations based on iterative partitioning that minimizes some distance metric between each observation and the centroid of some cluster

| classified identity \ true identity | Pb(II) | Cd(II) | Zn(II) | Cu(II) | Ni(II) | Co(II) | Fe(II) | Fe(III) | Cr(VI) | K⁺ | Mg²⁺ |
|---|---|---|---|---|---|---|---|---|---|---|---|
| Pb(II) | 111 | | 1 | | | | | 1 | 8 | | |
| Cd(II) | | 120 | | | | | | | | 3 | |
| Zn(II) | | | 61 | | | | | 8 | 3 | | |
| Cu(II) | | | 15 | 86 | | | | 2 | | 9 | |
| Ni(II) | | | | | 112 | | | | | | |
| Co(II) | | | | | | 120 | | | | | 30 |
| Fe(II) | | | | | | | 97 | | 19 | | |
| Fe(III) | | | 22 | | | | | 99 | | 24 | |
| Cr(VI) | 9 | | 43 | | 8 | | 13 | 1 | 89 | 1 | |
| K⁺ | | | | 12 | | | | 9 | 1 | 83 | |
| Mg²⁺ | | | | | | 10 | | | | | 90 |

**Figure 5.** Heavy metal ion and control cation classification by support vector machine (SVM) analysis for one trial at all scattering angles and particles sizes per concentration. Ions are organized from highest to lowest atomic mass. Data generated from 30 random samples.

**Table 1.** Classification comparison between support vector machine analysis (SVM), linear discriminant analysis (LDA, with posterior probabilities), K-means clustering and K-medians clustering. Rates are representative of each method with control cations included. Italicized numbers represent over or equal to 80% correct classifications.

| ion | SVM correctly classified (%) | LDA correctly classified (%) | posterior probability (%) | K-means correctly classified (%) | K-medians correctly classified (%) |
|---|---|---|---|---|---|
| Pb(II) | *92.5* | *83.3* | 86.1 | 50.0 | 75.0 |
| Cd(II) | *100.0* | *100.0* | 90.5 | *91.7* | *100.0* |
| Zn(II) | 50.8 | 66.7 | 99.0 | 50.0 | 50.0 |
| Cu(II) | 71.6 | 66.7 | 94.5 | 66.7 | 58.3 |
| Ni(II) | *93.3* | *91.7* | 92.4 | *83.3* | 75.0 |
| Co(II) | *100.0* | *100.0* | 91.2 | *100.0* | 75.0 |
| Fe(II) | *80.8* | *80.0* | 87.0 | *80.0* | *80.0* |
| Fe(III) | *82.5* | *83.3* | 70.7 | 50.0 | 50.0 |
| Cr(VI) | 74.2 | 66.7 | 64.2 | 41.7 | 33.3 |
| K⁺ | 69.1 | *83.3* | 78.2 | 50.0 | 58.3 |
| Mg²⁺ | 75.0 | 75.0 | 98.6 | 75.0 | *100.0* |

in the sample space. For K-means and K-medians clustering, these centroids are the cluster means and medians, respectively. Conversely, SVM serves as a classification method rather than a clustering method. From multiple distributions, SVM serves to identify which distribution a new data point lies within.

In these comparisons, SVM demonstrated the highest reproducible classification accuracy, most notably for Cd(II) and Co(II), followed by Pb(II), Ni(II), Fe(II) and Fe(III) (table 1). A similar trend could be observed for LDA—the highest classification accuracy for Cd(II) and Co(II), followed by Pb(II), Ni(II), Fe(II), Fe(III) and K⁺ (table 1).

## 3.3. Single-species concentration classification by SVM

In addition to classifying species, SVM analysis was used to classify the concentration of single-species heavy metal ions in solution. An SVM model was created such that a class was created for every heavy

**Table 2.** Positive classification rates of concentration classification by SVM for heavy metal ions, shown together with MCL's set by the US EPA [29–31]. Italicized numbers represent over or equal to 70% correct classifications.

| ion | **SVM**: correctly classified for both species and concentration (%) | | | |
| --- | --- | --- | --- | --- |
| | 1–1000 ppb | 10–1000 ppb | 100–1000 ppb | MCL by EPA (ppb) |
| Pb(II) | *100.0* | *100.0* | *100.0* | 15 |
| Cd(II) | 36.7 | 30.7 | 41.2 | 5 |
| Zn(II) | 40.4 | 53.9 | *80.9* | 5000 |
| Cu(II) | *83.0* | *77.3* | *100.0* | 1300 |
| Ni(II) | 51.1 | 59.0 | 52.6 | 100 |
| Co(II) | 62.7 | 50.2 | 44.2 | 2 |
| Fe(II) | 40.6 | 54.1 | *73.1* | 300 |
| Fe(III) | *75.3* | 67.1 | *74.3* | 300 |
| Cr(VI) | 51.2 | 51.6 | 27.5 | 100 |

metal ion and concentration. This resulted in 36 different potential classification classes from nine heavy metal ions ($K^+$ and $Mg^{2+}$ excluded) and four concentrations (1, 10, 100 and 1000 ppb). Results are summarized in table 2. Classification rates were summarized for three different ranges of concentrations, 1–1000, 10–1000 and 100–1000 ppb, which corresponded to the range of assays. Pb(II) showed 100% classification rates for all concentrations tested (1–1000 ppb), corresponding to 92.5% species classification rate shown in table 1. Fe(III) showed 75.3% classification rates for all concentrations tested (1–1000 ppb), again corresponding to 82.5% species classification rate (table 1). Fe(II) showed 73.1% classification rate only for high concentrations (100–1000 ppb), while the species classification rate was 80.8%. Cd(II), Co(II) and Ni(II), on the other hand, did not show high classification rates (30.7–62.7%), while the specifies classification data were quite high (100%, 93.3% and 100%, respectively). While Cd(II), Co(II) and Ni(II) were all correctly identified as Cd(II), Co(II) and Ni(II), respectively, the concentrations were frequently wrong, e.g. 1 ppb is frequently identified as 10 ppb, 10 ppb as 1 or 100 ppb, etc. On the other hand, decent concentration identification could be observed, 80.9%, for 100–1000 ppb Zn(II), despite its relatively poor species identification rate (50.8%). While species identification was successful for Pb(II), Cd(II), Ni(II), Co(II), Fe(II) and Fe(III), additional concentration identification was successful only for Pb(II), Fe(III), Fe(II) and Zn(II). Over 70% classification rates were observed for 1–1000 ppb Pb(II), 1–1000 ppb Fe(III), 100–1000 ppb Fe(II) and 100–1000 ppb Zn (II). Maximum contaminant levels (MCLs, set by US EPA) of these four species are 15 ppb, 300 ppb, 300 ppb and 5000 ppb, respectively [29–31]. As the lowest concentrations (analogous to limit of detection) in the concentration classifications are lower than MCLs, we can conclude that MCLs can be detected (with more than 70% classification rates) for these four species using our method.

## 3.4. Mixture identification by t-SNE

t-distributed stochastic neighbor embedding (t-SNE) plots were produced (figure 6) containing the three different mixtures described in the Material and method, namely (1) 15 ppb Pb(II) + 1300 ppb Cu(II) (EPA mandated action levels of pipe leaking—'Pipe 1'), (2) 5 ppb Pb(II) + 650 ppb Cu(II) (reduced tap monitoring levels of pipe leaking—'Pipe 2') and (3) 190 ppb Cd(II) + 110 ppb Ni(II) (subsurface water contamination in mine tailings—'Mine') and their respective single-species ions. More specifically, 100 ppb and 1000 ppb datasets of the individual single species were used. The aim of the plot was to use t-SNE dimension reduction to determine which of the composing ions had a more prominent influence on the measured scattering characteristics. This can be measured by determining the relative location of the mixture data to the change in concentration of the single-species data. Because of this aim, only the high concentration data (100 ppb and 1000 ppb) were used from single-species experiments.

The t-SNE plot displayed more prevalent interaction between the carboxylated polystyrene particles and Pb(II) for the 'Pipe 1' and 'Pipe 2' mixtures. This is indicated by the location of the 'Pipe 1' and

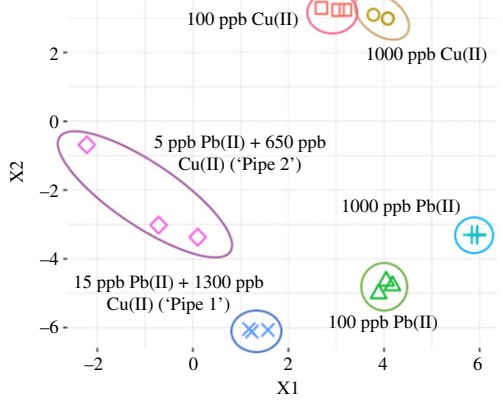

(a) t-SNE: Pb(II) and Cu(II) mixture and single species data

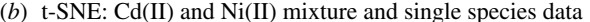

(b) t-SNE: Cd(II) and Ni(II) mixture and single species data

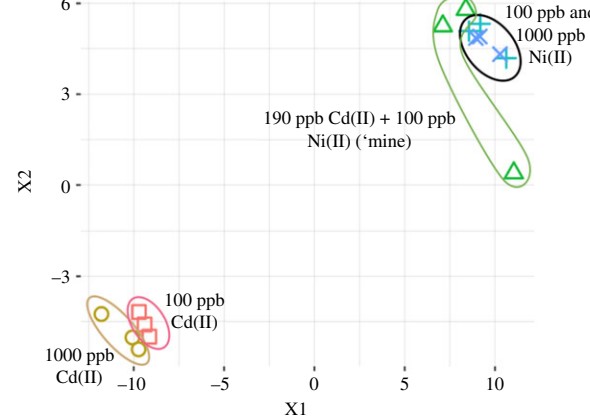

**Figure 6.** t-distributed stochastic neighbour embedding (t-SNE) dimension reduction applied to observations of mixtures and composing single-species ions using triplicate data of 0.75 μm, and 0.91 μm particles and corresponding angles. Results are shown for single species at 100 ppb and at 1000 ppb, as well as three mixtures: (1) 15 ppb Pb(II) + 1300 ppb Cu(II) ('Pipe 1'), (2) 5 ppb Pb(II) + 650 ppb Cu(II) ('Pipe 2') and (3) 190 ppb Cd(II) + 110 ppb Ni(II) ('Mine').

'Pipe 2' mixture groups with respect to the 100 ppb Pb(II) and 1000 ppb Pb(II) grouping compared to that of the 100 ppb Cu(II) and 1000 ppb Cu(II) grouping (figure 6a). It is also noted that as the relative ratio of the concentration of Pb(II) to Cu(II) increases from the 'Pipe 2' and 'Pipe 1' mixtures that the grouping of the 'Pipe 1' mixture grows nearer to the 100 ppb Pb(II) grouping. While SVM demonstrated about the same classification accuracy for Pb(II) and Cu(II), the scattering intensities with Pb(II) decreased more substantially than those with Cu(II) as shown in figure 3, which was attributed to the locations of 'Pipe 1' and 'Pipe 2' mixtures being closer to Pb(II) in the t-SNE plot. Similarly, the t-SNE plot displayed strong interaction between the carboxylated polystyrene particles and Ni(II) for the 'Mine' mixture as indicated by the location of 'Mine' grouping with respect to that of Ni(II) and Cu(II) individual groupings. Again, the scattering intensities with Ni(II) decreased more substantially that those with Cu(II) as shown in figure 3, despite the SVM classification accuracies being about the same for Ni(II) and Cu(II). To summarize, the relative concentration of heavy metal ions in a mixture can be estimated using t-SNE plots and a comprehensive dataset of individual ions at several concentrations.

To better identify multiple heavy metal ion species found in environmental water samples, it may be necessary to expand this method to mixtures of more than three heavy metal ion species. However, the amount of data that would need to be collected to develop such a multiplexed model would be extraordinarily high, requiring extensive labour and resources. For most groundwater and drinking water samples, however, as few as one or two major heavy metal ion species are often observed, as we have simulated in this study (Pb(II) and Cu(II) in pipe leaching and Cd(II) and Ni(II) in subsurface mine tailing models). This is true for other types of groundwater or drinking water samples. Taghipour *et al.* [32] reported the heavy metal contamination of groundwater used for irrigation in Tabriz, Iran, and found that the average Zn(II) and Cu(II) concentrations were 49 ppb

(maximum 204 ppb) and 16 ppb (maximum 37 ppb), respectively, while all other heavy metal ion concentrations were less than 6.6 ppb (average was 3.8 ppb). Rahmanian *et al.* [33] reported on drinking water contamination in Perak, Malaysia, and found that the average Mg and Fe concentrations were 119–512 ppb and 12–67 ppb, respectively, while all other heavy metal ion concentrations were less than 7 ppb (less than 2 ppb for 78% of data).

# 4. Conclusion

In conclusion, support vector machine and discriminant analyses indicate that there is a distinguishable difference in the scattering characteristics between different heavy metal ions and control cations. Among these, the most characteristically distinguishable heavy metal ions are Pb(II), Cd(II), Ni(II) and Co(II), followed by Fe(II), and Fe(III). These findings suggest that the sorption of heavy metals on simple carboxylated polystyrene microparticles differs somewhat between species, particularly for Pb(II), Cd(II), Ni(II) and Co(II), as well as for Fe(II) and Fe(III). These differences may be increasingly distinguishable between an array of particle sizes and detection angles, even among relatively small data sets. With increasing data sets, measurements of additional parameters (e.g. sample conductivity) and a more complex chemical binding group, improved classification of these heavy metal ion species may be possible in samples with diverse mixtures. In addition, it may also be possible to identify heavy metal ion species in real water systems, i.e. with co-existing cations.

Data accessibility. Data available from the Dryad Digital Repository: https://doi.org/10.5061/dryad.62n8p0q.
Authors' contributions. K.E.K. and J.Y.Y. conceived the original concept and designed the experiments. K.E.K. and C.M.J. prepared all samples/reagents and collected all scattering data, with consultation from J.Y.Y. K.E.K., C.M.J. and N.L. analysed all scattering data using various statistical inference methods, with consultation from L.A. All authors analysed the classification results and wrote the manuscript.
Competing interests. We have no competing interests.
Funding. K.E.K. acknowledges support from the U.S. National Science Foundation (NSF) Graduate Research Fellowship Program (GRFP) under grant no. DGE-1143953. Any opinion, findings, and conclusions or recommendations expressed in this material are those of the author(s) and do not necessarily reflect the views of the NSF. C.M.J. and J.Y.Y. acknowledge partial support from the U.S. National Institutes of Health – National Institute of Environmental Health Sciences (NIH-NIEHS), award no. R25ES025494 and from the Western Alliance to Expand Student Opportunities (WAESO) at Arizona State University. N.L. and L.A. acknowledge partial support from the U.S. National Institutes of Health – National Institute of General Medical Sciences (NIH-NIGMS), award no. T32GM084905. Additional support was provided by Korea Institute of Ocean Science and Technology (KIOST) awarded to J.Y.Y.

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
