## [Reviewer comments · Royal Society Open Science]

Review History

RSOS-190001.R0 (Original submission)

Review form: Reviewer 1

Is the manuscript scientifically sound in its present form?

No

Are the interpretations and conclusions justified by the results?

No

Is the language acceptable?

Yes

Is it clear how to access all supporting data?

Yes

Do you have any ethical concerns with this paper?

No

Have you any concerns about statistical analyses in this paper?

I do not feel qualified to assess the statistics

Recommendation?

Major revision is needed (please make suggestions in comments)

Comments to the Author(s)

This manuscript propose a promising method for classifying heavy metals in aqueous solution using Mie scattering measurements associated with each ion's characteristic activity with different sizes of microparticles linked with metal ions. This study mainly was aimed to probe how well statistical inference methods could classify the heavy metal ion species in the water samples using non-specific assays. This study is interesting for identify heavy metal ions in water. However, some problems need to be resolved or addressed appropriately before publication.

(1) Fig. 1 needs to be explained in more detail. For example, How can we obtain the dotted plot in the left of Fig. 1? and how to distinguish whether the microparticles is aggregated.

(2) It is suggested that the authors may explain why select K^+ and Mg^{2+} as the control cations.

(3) In this study, the mixed species solutions just contain two metals. However, there are many heavy metals generally coexisting in natural water. Does the method proposed in this study work for the complex aqueous solution?

(4) It is better to provide field application examples to strengthen the feasibility and reliability of the technique proposed in this study.

Review form: Reviewer 2

Is the manuscript scientifically sound in its present form?

Yes

Are the interpretations and conclusions justified by the results?

Yes

Is the language acceptable?

Yes

Is it clear how to access all supporting data?

Yes

Do you have any ethical concerns with this paper?

No

Have you any concerns about statistical analyses in this paper?

I do not feel qualified to assess the statistics

Recommendation?

Accept as is

Comments to the Author(s)

This paper offers the possibility to simplify heavy metal analysis in environmental samples. It has been developed a way to classify heavy metal species in water samples using non-specific assays.

At this moment the paper shows limitations related to sensitivity/specificity in the assays, but they demonstrated the valuation of a methodology that can be improved through the generation and analysis of a bigger set of data. Finally, including more variables in the assays, maybe they could fill the requirements of EPA for complex samples, like environmental samples of contaminated water. This is a paper that the journal Royal Society Open Science can publish.

Decision letter (RSOS-190001.R0)

01-Mar-2019

Dear Dr Yoon:

Title: Mie scattering and microparticle based characterization of heavy metal ions and classification by statistical inference methods
Manuscript ID: RSOS-190001

The editor assigned to your manuscript has now received comments from reviewers. We would like you to revise your paper in accordance with the referee and Subject Editor suggestions which can be found below (not including confidential reports to the Editor). Please note this decision does not guarantee eventual acceptance.

Please submit your revised paper before 24-Mar-2019. Please note that the revision deadline will expire at 00.00am on this date. If we do not hear from you within this time then it will be assumed that the paper has been withdrawn. In exceptional circumstances, extensions may be possible if agreed with the Editorial Office in advance. We do not allow multiple rounds of revision so we urge you to make every effort to fully address all of the comments at this stage. If deemed necessary by the Editors, your manuscript will be sent back to one or more of the original reviewers for assessment. If the original reviewers are not available we may invite new reviewers.

RSC Associate Editor:
Comments to the Author:
(There are no comments.)

RSC Subject Editor:
Comments to the Author:
(There are no comments.)

Reviewers' Comments to Author:
Reviewer: 1

Comments to the Author(s)

This manuscript propose a promising method for classifying heavy metals in aqueous solution using Mie scattering measurements associated with each ion's characteristic activity with different sizes of microparticles linked with metal ions. This study mainly was aimed to probe how well statistical inference methods could classify the heavy metal ion species in the water samples using non-specific assays. This study is interesting for identify heavy metal ions in water. However, some problems need to be resolved or addressed appropriately before publication.

- (1) Fig. 1 needs to be explained in more detail. For example, How can we obtain the dotted plot in the left of Fig. 1? and how to distinguish whether the microparticles is aggregated.
- (2) It is suggested that the authors may explain why select K^+ and Mg^{2+} as the control cations.
- (3) In this study, the mixed species solutions just contain two metals. However, there are many heavy metals generally coexisting in natural water. Does the method proposed in this study work for the complex aqueous solution?
- (4) It is better to provide field application examples to strengthen the feasibility and reliability of the technique proposed in this study.

Reviewer: 2

Comments to the Author(s)

This paper offers the possibility to simplify heavy metal analysis in environmental samples. It has been developed a way to classify heavy metal species in water samples using non-specific assays. At this moment the paper shows limitations related to sensitivity/specificity in the assays, but they demonstrated the valuation of a methodology that can be improved through the generation and analysis of a bigger set of data. Finally, including more variables in the assays, maybe they could fill the requirements of EPA for complex samples, like environmental samples of contaminated water. This is a paper that the journal Royal Society Open Science can publish.

Author's Response to Decision Letter for (RSOS-190001.R0)

See Appendix A.

Decision letter (RSOS-190001.R1)

05-Apr-2019

Dear Dr Yoon:

Title: Mie scattering and microparticle based characterization of heavy metal ions and classification by statistical inference methods
Manuscript ID: RSOS-190001.R1

It is a pleasure to accept your manuscript in its current form for publication in Royal Society Open Science. The chemistry content of Royal Society Open Science is published in collaboration with the Royal Society of Chemistry.

RSC Associate Editor
Comments to the Author:
(There are no comments.)

Reviewer(s)' Comments to Author:

Appendix A

RESPONSE LETTER

Title: Mie scattering and microparticle based characterization of heavy metal ions and classification by statistical inference methods

Manuscript ID: RSOS-190001

Dear Editor and Reviewers,

We appreciate the constructive feedback and evaluation of our manuscript. In the following text below, we address the reviewers' points of concern.

REVIEWER 1

Comment: *This manuscript propose a promising method for classifying heavy metals in aqueous solution using Mie scattering measurements associated with each ion's characteristic activity with different sizes of microparticles linked with metal ions. This study mainly was aimed to probe how well statistical inference methods could classify the heavy metal ion species in the water samples using non-specific assays. This study is interesting for identify heavy metal ions in water. However, some problems need to be resolved or addressed appropriately before publication.*

Comment 1: *Fig. 1 needs to be explained in more detail. For example, how can we obtain the dotted plot in the left of Fig. 1? And how to distinguish whether the microparticles is aggregated.*

Response: I believe the reviewer is referring the "dotted plot" in the left of Fig. 1 as the series of holes in the 3D printed device. A detection optical fiber is inserted into one of these holes to collect scattering intensity at a specific angle. The angle showing the maximum scatter intensity shifts to a different angle upon particle aggregation, which is used for identifying heavy metal ions. The "dotted plots" on the right of Fig. 1 are images of the single and aggregated particles: the upper images were modified from grayscale to binary to identify particles, and the lower are false color images in which single particles (0.91 μm in diameter) are distinguished from aggregated particles.

Figure 1 caption is modified: "Fig. 1. (left) Mie scattering intensity is collected at a specific detection angle through inserting a detection optical fiber into one of the holes in the 3D printed device. The angle showing the maximum Mie scatter intensity shifts to a different angle upon particle aggregation, which can be used for identifying heavy metal ions. (right) Carboxylated polystyrene microparticles characteristically interact with heavy metal ions (Me^{2+}). This leads to particle aggregation, e.g. effective particle size changes from singlets (green) to aggregates (red), shown via microscopy with 0.91 μm particles in the presence of 1 and 10 ppb Cr(VI)."

Comment 2: *It is suggested that the authors may explain why select K^+ and Mg^{2+} as the control cations.*

Response: They are control cations – K^+ was chosen from group 1 and Mg^{2+} from group 2 in the periodic table, considering their atomic mass and environmental availability. As we are not quantifying those cations and simply using them as controls, we have chosen one cation from each group.

Added to section 2.4: “... Assay samples containing single heavy metal species or single reference cation species (K^+ or Mg^{2+} ; chosen from groups 1 and 2 from the periodic table considering their atomic mass and environmental availability) were prepared separately with each PS-COOH sample (D = 0.91 μm , 0.75 μm , and 0.40 μm) in a 50:50 ratio. ...”

Comment 3: *In this study, the mixed species solutions just contain two metals. However, there are many heavy metals generally coexisting in natural water. Does the method proposed in this study work for the complex aqueous solution?*

Response: As addressed at the end of Introduction, single species identification/quantification was the main aim of this work, while we have also demonstrated preliminary assays of binary heavy metal mixtures. In fact, binary mixtures of heavy metals are predominantly found in groundwater or drinking water contamination scenarios. In pipe leaching into drinking water case, Pb(II) and Cu(II) were the two major heavy metal ions as reported by references 13 and 14. In subsurface water contamination in mine tailings, Cd(II) and Ni(II) were the two major heavy metal ions as reported by references 16 and 17. Other examples of such water contamination by binary heavy metal ions are added in this revision.

Added to the end of section 3.4: “To better identify multiple heavy metal ion species found in environmental water samples, it may be necessary to expand this method to mixtures of more than three heavy metal ion species. However, the amount of data that would need to be collected to develop such a multiplexed model would be extraordinarily high, requiring extensive labor and resources. For most groundwater and drinking water samples, however, as few as one or two major heavy metal ion species are often observed, as we have simulated in this study (Pb(II) and Cu(II) in pipe leaching and Cd(II) and Ni(II) in subsurface mine tailing models). This is true for other types of groundwater or drinking water samples. Taghipour et al.³² reported the heavy metal contamination of groundwater used for irrigation in Tabriz, Iran, and found that the average Zn(II) and Cu(II) concentrations were 49 ppb (maximum 204 ppb) and 16 ppb (maximum 37 ppb), respectively, while all other heavy metal ion concentrations were < 6.6 ppb (average was 3.8 ppb). Rahmanian et al.³³ reported on drinking water contamination in Perak, Malaysia, and found that the average Mg and Fe concentrations were 119-512 ppb and 12-67 ppb, respectively, while all other heavy metal ion concentrations were < 7 ppb (< 2 ppb for 78% of data).”

Two new references were added:

32. H. Taghipour, M. Mosaferi, M. Pourakbar and F. Armanfar, *Health Promot. Perspect.*, 2012, **2**, 205-210.

33. N. Rahmanian, S. H. B. Ali, M. Homayoonfard, N. J. Ali, M. Rehan, Y. Sadeef and A. S. Nizami, *J. Chem.* 2015, **2015**, 716125.

Comment 4: *It is better to provide field application examples to strengthen the feasibility and reliability of the technique proposed in this study.*

Response: We recognize that there are inherent physical limitations to the sensitivity/specificity of the carboxylated particle aggregation assay of heavy metal ions. However, the main point of the paper was an exploration of how well statistical inference methods could classify heavy metal ion species in the water samples using non-specific assays. These methods work better with larger datasets, so there are limitations to this paper in that regard. However, we aimed to show in this work that reasonable detection could be achieved under a set of controlled circumstances even with a relatively limited amount of data (given the number of total ions and combinations we evaluated).

The system proposed in the paper could be applicable in real water systems if an extensive dataset encompassing the aggregation of the PS-COOH particles and heavy metal ions in an environment with other chemical species was recorded through statistical classification methods outlined in the paper. Thus, it is not our primary focus for this paper to explore the effects of other chemical species when the paper's purpose is to show how polystyrene particle aggregation, Mie scattering measurements, and statistical classification can be used to make a sensitive and specific environmental sensor.

To adapt this method for more complicated field water samples, e.g. river water, a huge amount of data would need to be collected with all possible combinations of heavy metal ion species and their concentrations. Such effort would require a large-scale, multi-institutional collaboration, which we hope to garner through this work.

REVIEWER 2

Comments. *This paper offers the possibility to simplify heavy metal analysis in environmental samples. It has been developed a way to classify heavy metal species in water samples using non-specific assays. At this moment the paper shows limitations related to sensitivity/specificity in the assays, but they demonstrated the valuation of a methodology that can be improved through the generation and analysis of a bigger set of data. Finally, including more variables in the assays, maybe they could fill the requirements of EPA for complex samples, like environmental samples of contaminated water. This is a paper that the journal Royal Society Open Science can publish.*

Response: Thank you very much for the positive comments. Please refer to our responses to the reviewer 1 regarding the current limitations and potential applications to complex samples.